# Protein and Amino Acid Supplementation Among Recreational Gym Goers and Associated Factors—An Exploratory Study

**DOI:** 10.3390/jfmk10030248

**Published:** 2025-06-28

**Authors:** Sandor-Richard Nagy, Magdalena Mititelu, Violeta Popovici, Mihaela Gabriela Bontea, Annamaria Pallag, Tünde Jurca

**Affiliations:** 1Doctoral School of Biomedical Sciences, Faculty of Medicine and Pharmacy, University of Oradea, 410073 Oradea, Romania; nagy.sandorrichard@student.uoradea.ro (S.-R.N.); apallag@uoradea.ro (A.P.); tjurca@uoradea.ro (T.J.); 2International Fitness and Bodybuilding Federation, 28232 Madrid, Spain; 3Department of Clinical Laboratory and Food Safety, Faculty of Pharmacy, “Carol Davila” University of Medicine and Pharmacy, 020956 Bucharest, Romania; magdalena.mititelu@umfcd.ro; 4Center for Mountain Economics, “Costin C. Kiriţescu” National Institute of Economic Research, Romanian Academy, 725700 Vatra-Dornei, Romania; 5Department of Psychoneuroscience and Recovery, Faculty of Medicine and Pharmacy, University of Oradea, 410073 Oradea, Romania; 6Department of Pharmacy, Faculty of Medicine and Pharmacy, University of Oradea, 410073 Oradea, Romania

**Keywords:** preventive health, recreational gym goers, daily diet type, nutritional supplements, whey protein, creatine, L-carnitine, physical enhancement, potential side effects

## Abstract

**Objective:** The present study investigated the relationship between protein and amino acid supplementation and various associated aspects among recreational gym goers at 2 gymnasiums in Oradea (Romania). **Methods:** A total of 165 gym goers (110 men and 55 women, most of them 18–30 years old) with high educational levels were included in the present study, which was conducted as face-to-face interviews. **Results:** Participants were divided into 4 groups: protein supplement users (PSUs, 42/165), creatine supplement users (CSUs, 38/165), L-carnitine supplement users (LcSUs, 37/165), and protein + creatine + L-carnitine supplement users (PCLcSUs, 48/165). Most consumers were young (18–30 years) and preferred the triple combination. Females consumed PS and CS (38.2% and 34.5%, respectively), while the most-used NSs by males were PCLcS (36.4%) and LcS (27.3%). Obese gym goers opted for LcS consumption (r = 0.999, *p* < 0.05). Creatine and L-carnitine were consumed for force training (65.79 and 62.16%), while PCLcS and PS were used in cardio + force and force training in equal measures (42.86 and 47.92%, respectively). Most PSUs were gym goers for 7–12 months and more than 1 year (r = 0.999 and r = 0.952, respectively, *p* < 0.05), while PCLcSUs had a training frequency of at least 5 times a week (r = 0.968, *p* < 0.05). Muscle mass growth was the primary training focus for all NS users (57.89%), followed by muscular tonus (40.54%, *p* < 0.05). Almost 30% of one-only NS users reported various side effects, whereas all PCLcSUs claimed side effects (*p* < 0.05). **Conclusions:** Age and gender were key factors in diet type, training type, frequency, duration, scope, NS type, and dose intake. The frequency of side effects substantially depended on the kind of NS and the dose consumed. The present study’s results highlight the need for health professionals’ advice and monitoring in personalized diets and protein and amino acid supplementation in recreational gym goers.

## 1. Introduction

Both nutrition and physical activity are fundamental pillars of good health, offering many benefits that impact nearly every aspect of our well-being. They work synergistically to promote a healthier and longer life. Due to environmental concerns raised by the food industry, the World Health Organization (WHO) supports sustainable diets, defined as dietary patterns that underline all dimensions of individuals’ health and well-being [1,2]. Regular physical activity, encompassing any body movement that expends energy, offers a multitude of physical and mental health advantages, including improved cardiovascular health, help in managing weight, building and maintaining strong muscles and bones, increased energy levels, improved mood, diminished stress, enhanced cognitive function, facilitating recovery after various diseases, reduced risk of chronic ones, and ameliorating patient condition in various chronic illnesses [3].

Numerous people embrace a lifestyle that incorporates a balanced diet and regular exercise as a key to a healthier life. Most use gyms and fitness facilities for general health, weight loss, and fitness improvement and are known as recreational gym goers [4]. They incorporate gym attendance and exercise into their lives for individual benefits and enjoyment without the pressures and demands of professional athletes [5]. Currently, recreational gym goers excessively focus on improving fitness and achieving aesthetic goals, such as building muscle or losing fat, using various nutritional supplements (NSs). Although the use of NSs by professional athletes and their benefits have been extensively studied, the literature on recreational gym goers is limited [6].

### 1.1. Literature Review

#### 1.1.1. Protein and Amino Acid Supplementation—Benefits and Daily Doses

Protein supplements, particularly whey protein (WP), are among the most popular supplements due to their role in muscle synthesis and recovery [7]. The Institute of Medicine’s recommended dietary allowance (RDA) for protein is 0.8 g/kg/day for the entire adult population [8], while the Acceptable Macronutrient Distribution Range (AMDR) indicates a daily protein intake of 1.05–3.67 g per kilogram of body weight for adults aged 18 years and older [9]. The International Society of Sports Nutrition showed that for most people who exercise, an overall protein intake in the range of 1.4–2.0 g protein/kg body weight/day (g/kg/d) is adequate for gaining and maintaining muscle mass through a positive muscle protein balance. More protein (2.3–3.1 g/kg/d) could be required for resistance-trained subjects to optimize lean body mass retention during hypocaloric periods. Higher protein intake (>3.0 g/kg/d) may improve body composition in resistance-trained individuals (encourage loss of fat mass) [10].

Whey protein significantly impacts nutritional supplements for athletes, as it contains around 50% of essential amino acids (EAAs) and approximately 26% of branched-chain amino acids (BCAAs). The amino acid composition of WP exhibits a similar pattern to that of human skeletal muscle, allowing for faster absorption compared to other protein sources [11]. It can stimulate skeletal muscles, reduce fatigue, enhance muscle protein synthesis (MPS), and slightly inhibit muscle protein breakdown (MPB) [12]. Recreational exercisers widely practice resistance training (RT) combined with WP supplementation to increase the muscle mass and strength that RT induces [13]. A daily dose of 20 to 25 g of WP yields the intended advantages. However, exceeding 40 g may cause adverse effects [14].

Creatine is a well-documented ergogenic aid that facilitates ATP production by enhancing muscle strength, lean body mass, and recovery [15]. It is a guanidine compound with both exogenous and endogenous sources that is synthesized in the kidneys, liver, and pancreas from three amino acids: glycine, methionine, and arginine [16,17]. Creatine monohydrate (CM) supplementation has been consistently reported in the literature to increase phosphagen levels in muscle, improve performance during repetitive high-intensity exercise, and promote significant training adaptations [18]. It is a stable form of creatine that is not significantly degraded during the digestive process and is either taken up by muscle or eliminated in urine. Despite its widespread use worldwide, no clinically significant adverse effects have been reported from CM supplementation; short- and long-term supplementation is safe and well-tolerated in healthy individuals and several patient populations. The regulatory status of CM is not well established. Currently, it is the only form of creatine officially approved in key markets, including the USA, Canada, the European Union, and Republic of Korea [18,19,20,21,22,23,24,25,26]. The authors of a previous study concluded that a dose of up to 3 g/day is both safe and effective in terms of changes in strength and body composition [27]. A recent study examined the effects of whey protein and creatine supplementation compared to WP consumption alone on body composition and performance variables in 17 resistance-trained young women [28]. After 8 weeks of training, Wilborn et al. observed that performance in all groups increased; however, no significant differences were recorded between the WP and WP + CM supplementation groups [28]. Similar results were reported by Collins et al. in RT of elderly individuals with frailty [29].

L-carnitine, another widely used supplement, is valued for facilitating the metabolism of fatty acids and energy production within mitochondria [30]. Increasing L-carnitine intake through supplementation can enhance fat oxidation, significantly reducing body fat reserves [31]. Several studies reported that LcS increases exercise performance, improves recovery, and reduces oxidative stress [32,33]. Although naturally present in animal-based foods, its supplementation is often necessary for individuals with higher metabolic demands or specific treatments and dietary restrictions [34,35,36,37,38,39,40,41]. L-carnitine tartrate supplementation has a beneficial effect on markers of post-exercise metabolic stress and muscle damage. Spiering et al. demonstrated that 1–2 g L-carnitine effectively mitigated various markers of metabolic stress and muscle soreness in athletes [42] while 3–4 g taken before physical exercise prolonged exhaustion [43]. Evans et al. reported the benefits of L-carnitine combined with creatine and leucine on functional muscle strength in healthy older adults [44].

#### 1.1.2. Protein and Amino Acid Supplements: Potential Side Effects and Nutrivigilance

Protein supplements are generally safe for most people when used as recommended, but they can cause side effects, especially when overused or in individuals with underlying health conditions [45]. The well-known side effects are digestive issues (abdominal pain, gas, diarrhea), kidney strain [46,47,48], liver damage (when the liver is previously affected) [49,50], allergic reactions [51], nutrient imbalance, heavy metal contamination [52], and weight gain. Several less common side effects could be hormonal imbalance due to soy protein, acne [53,54], skin issues due to WP, and bone demineralization [55].

Creatine supplementation is safe for healthy individuals at recommended dosages [56]. However, like any supplement, it may cause side effects in some users. Kidney and liver damage are potential side effects supported by scientific literature [57,58,59,60].

L-carnitine, in higher doses than 3 g/daily and prolonged treatment, can induce liver and kidney damage [61], digestive discomfort, fishy body odor, rash, seizure, and high blood pressure [62].

For safeguarding consumer health by identifying potential risks associated with nutritional supplements, nutrivigilance is the science and practice of monitoring, detecting, assessing, and preventing adverse effects related to consuming food products, particularly dietary supplements, fortified foods, novel foods, and foods for a specific population [63]. Nutrivigilance plays a considerable role in ensuring the post-market safety of nutritional supplements [64]. By systematically collecting and analyzing data on adverse effects, authorities can identify potential risks, inform regulatory decisions, and protect public health [65].

While some countries have established national nutrivigilance systems [66], there is a lack of harmonization across the European Union [67]. This disparity leads to inconsistencies in monitoring and managing the safety of food supplements and related products [68]. Experts emphasize the necessity for a coordinated European nutrivigilance system to facilitate data sharing, risk assessment, and regulatory actions, thereby enhancing consumer protection across member states [69]. Establishing and harmonizing nutrivigilance systems are essential to achieving comprehensive consumer safety in food supplements and related products [70].

#### 1.1.3. Protein and Amino Acid Supplement Consumption in Recreational Gym-Goers

The continuously increasing prevalence of the consumption of various NSs without a healthcare professional’s recommendation can be explained by the widespread belief that their regular intake can improve consumer health [71,72]. Recreational gym goers use NSs more than professional athletes [73]. Unexpectantly, a recent study reported that, in Brazil, most consumers of sports dietary supplements are physically inactive [74].

While individual preferences and objectives vary, some supplements are more popular. Athletes consume the most vitamins (75.3%), recreational gym goers prefer protein alone (30.8%) or in combination with creatine (12.2%) [75]. Thomas et al. observed that protein supplement consumption is linked to time spent exercising and high-protein-content foods [76]. Several authors suggest that athletes need extra proteins in their diet as food or as supplements, but regular gym goers do not need these extra supplements [77]. However, physicians and nutritionists have been poorly consulted [78].

Then, recreational gym goers’ protein consumption is based on their own beliefs [78,79,80]. Creatine and protein are in the top 5 of the most used NSs [81]. Frequently consumed by gym goers are protein powders (59.17%), followed by creatine (41.28%), and L-carnitine (5.05%) [82]. Most recreational gym goers prefer protein associated with creatine and amino acids (48.8%) or protein and creatine alone (6.4%) [78]. Peeling et al. included creatine in the group of established performance supplements. At the same time, L-carnitine was considered an equivocal one, with less clear evidence for its potential to enhance athletic performance [83]. Numerous questionnaire-based studies have investigated protein and amino acid supplementation among recreational gym goers worldwide, as well as associated factors. Most of them analyzed the most-used supplements from various categories. They correlated their use with some aspects, such as age, sex, diet, bad habits, body weight, education, income, training type, frequency, scope, etc. [84,85,86,87,88,89,90,91,92,93,94,95,96,97,98].

Several studies have explored the increasing trend of non-steroidal anti-inflammatory drug (NSAID) consumption without medical advice among Romanian athletes. Ionescu et al. described a new model of analytical and prospective tools that explore nutritional supplement consumption and their positive or harmful effects [99]. Another research team used an online questionnaire to provide original information on dietary supplement use, type, effects, and source of purchase among healthy residents in Târgu Mureș, Romania [100]. Another study highlighted the necessity of introducing nutrivigilance as a habitual practice and activity of all authorities and actors in the Romanian dietary supplement market [101].

##### Hypotheses

The literature review led to the following hypotheses:Recreational gym goers commonly use protein and amino acid supplements alone or in combination [78,94,102];Most of them are men who are young and highly educated [75,79,80,103,104];Various socio-demographic and training factors influence gym goers’ protein and amino acid supplement consumption [105,106,107,108,109,110];Protein and amino acid supplements may have side effects claimed by consumers depending on NS type and daily dose [62,111,112].

##### The Aim of the Present Study

In this context, the present study examined the consumption of protein, creatine, and L-carnitine, both individually and in combination, among recreational gym goers in Northeastern Romania. It aimed to analyze the reasons for using each nutritional supplement and the significant factors that influence its use. Based on the above-mentioned recent studies, a similar complex and extensive analysis of protein and amino acid supplementation in a heterogeneous group of Romanian recreational gym goers has not yet been performed.

## 2. Materials and Methods

### 2.1. Study Design

The present study followed the Declaration of Helsinki regulations [113]. The protocol was approved by the Ethical Committee of the Faculty of Medicine and Pharmacy, University of Oradea, Romania (No. 21/25 February 2021), and the Gym Committee (No 7/3 October 2022).

This cross-sectional study was conducted as face-to-face interviews at two popular gymnasiums in Oradea, Romania, from 15 January to 15 December 2024. Participants were regular gym goers in these 2 gyms, males and females aged 18–60 years old, Romanian language speakers, and residents of the Oradea Metropolitan Area [114]. The inclusion and exclusion criteria are displayed in Table 1.

Two qualified coaches conducted a rigorous check and selection process for potential participants at both gym locations. Only 165 fulfilled the inclusion criteria, 55 females and 110 males. They revealed their preference for protein, creatine, and L-carnitine supplements; they had regular training sessions at least twice a week at one of the gym locations. All participants were differentiated by supplement type: 42/165 were protein supplement users (PSU), 38/35 were creatine supplement users (CSU), 37/165 were L-carnitine supplement users (37/165), and 48/165 preferred to consume them in combination—protein, creatine, and L-carnitine supplement users (PCLcSUs, 48/165).

### 2.2. Method

Participants received brief instructions about the purpose and nature of the present research. Written informed consent was obtained from each participant who voluntarily participated in the present study. The data collection tool was a face-to-face interview based on a complex questionnaire (available in the Appendix A) adapted from previous works [6,82,116,117]. The participants were assured that the researchers would maintain their privacy and keep the data collected private. Measurements of body weight and height were made for all individuals before the interview, and the body mass index (BMI) was calculated under WHO guidelines using the Quetelet equation: body weight (kg)/height^2^ (m^2^) [118].

The questionnaire’s first section focused on demographic and socio-economic information. Age, gender, education level, occupation, and monthly income were included in the socio-demographic data-related inquiries. The participants were asked about their type and degree of activity at work, classifying them as follows: completely sedentary in office jobs, physically active jobs, and a combination of both.

The second section of the questionnaire investigated the participants’ lifestyles and healthcare levels (body weight, unhealthy habits, diet type, daily calorie consumption, daily protein consumption, and frequency of daily meals).

The third section contained data about gym workout routines. It included detailed information about how long it had been since the participants started working out at the gym, how many days a week they work out there, how many hours/minutes they spend in the gym, what kinds of exercises they do there—such as cardio exercises, strength training, and a mix of both exercises—and the scope of gym training.

The fourth section of the questionnaire examined the participants’ awareness of NSs, their motivation for using them, administration details, benefits, and any claimed side effects after using them in conjunction with gym practice.

### 2.3. Statistical Analysis

The reliability analysis of the internal model of XLSTAT Premium v.2024.4.2.1426 by Lumivero (Denver, CO, USA) investigated the questionnaire, examining the intercorrelation between all questions. Cronbach’s alpha index and Guttman L1–L6 coefficients were calculated [119].

Descriptive statistics were computed to summarize and analyze the dataset’s central tendency, dispersion, and distribution, providing essential insights into the study variables. This analysis was conducted using XLSTAT Premium 2024 v. 2024.4.2.1426 and followed methodologies outlined in previous research studies [120,121]. Data are expressed as frequency (number) and relative frequency (percentage).

Furthermore, principal component analysis (PCA) was performed using Pearson’s correlation to detect the relationships among variables [122]. The modifiable factor was NS consumption, while the influential ones were all aspects investigated in the face-to-face questionnaire.

Finally, the chi-square independence test assessed the significance of each variable parameter’s influence on the risk of potential side effects, calculating *p*-values.

The statistical significance was set at *p*-value < 0.05, indicating that results below this threshold were considered statistically significant, aligning with standard health and nutrition research practices.

All statistical tools used ensured robust and reliable data analysis, facilitating the accurate interpretation of relationships between variables.

## 3. Results

### 3.1. Reliability Analysis

The Cronbach’s alpha index value was 0.947, and the Guttman L1–L6 coefficients were 0.910–1.000. The correlation matrix, covariance matrix, and high coefficients revealed that all questions were significantly intercorrelated. The data obtained confirmed the questionnaire’s substantial reliability and appreciable internal consistency, thus confirming its high quality. More details are available in the Appendix A.

### 3.2. Socio-Demographic Data of Participants

The present study enrolled 165 recreational gym goers who preferred various NSs. Of the participants, 33.33% (55/165) were women and 66.67% (110/165) were men (*p* < 0.05). Over 50% (87/165, 52.73%) were 18–30 years old, and 2.42% (4/165) were 50–60 years old (*p* < 0.05). Similar percentages (23.64% and 21.21%, respectively) belonged to the 30–40 and 41–50 age groups.

Most participants (135/165) had a substantial educational level: university studies (105/165, 63.64%) and master’s/doctorate (30/165, 18.18%), while only 18.18% (30/165) had a high school education (*p* < 0.05). Monthly income varied from <2000 RON (3/165, 1.82%) to >6000 RON (43/165, 26.06%) (*p* < 0.05). Most respondents had 2000–4000 RON (75/165, 45.45%), followed by those with 4001–6000 RON (44/165, 26.67%) (*p* < 0.05) (Table 2).

The daily work (DW) regimen involved >8 h for most of them (91/165, 55.15%), followed, in decreasing order, by those with 4–8 h (69/165, 41.82%) and 1–4 h (5/165, 3.03%) (*p* < 0.05). Their daily work consisted of physical activity (44/165, 26.67%) and office work (54/165, 32.73%); 40.61% of participants (67/165) had combined work (physical activity and office work) (*p* < 0.05) (Table 2).

### 3.3. Lifestyle Patterns and Self-Care Awareness

The following questions investigated the participants’ lifestyles: unhealthy habits, diet type, daily meal frequency and nutritional content, body weight type, and illness history. All data are displayed in Table 3.

Most gym goer participants were non-smokers (116/165, 70.3%), while only 29.70% (*p* < 0.05) declared that they were smokers (daily and occasionally, in similar percentages: 15.15% vs. 14.55%, *p* > 0.05). Only 19.39% of respondents stated they were not alcohol consumers (32/165), while most of them consumed alcohol (80.61%, *p* < 0.05). Over 60% of respondents (107/165, 64.85%) had a balanced diet (Table 2). Extreme diets (hyperprotein, vegetarian, and low-carb) had significantly lower incidences (19.39% vs. 9.7% vs. 6.06%, *p* < 0.05). Most respondents had 3 meals/day (49.49%, 81/165) or 3 meals + 2 snacks (31.52%, 52/165) (*p* < 0.05). Several gym goers had 2 meals/daily (18/165, 10.91%), while 8.48% (14/165) preferred intermittent fasting (Table 2). The daily calorie (DC) rate ranged between 1000–1500 Cal and >3500 Cal. Over 50% (88/175, 53.33%) consumed 1501–2500 Cal/day (Table 2). Around 5/165 (3.03%) participants did not know their DC consumption; most were men (4/165) vs. women (1/154) (*p* < 0.05).

Daily protein (DP) intake varied between <50 g (1/165, 0.61%) and >250 g (7/165, 4.24%) (*p* < 0.05). Most respondents consumed 50–150 g of protein daily (79/165, 47.88%) and had a normal weight (83/165, 50.30%) or were overweight (76/165, 46.06%). Substantial differences were recorded between women and men in the overweight category (14.55% vs. 61.82%, *p* < 0.05). Only 2 participants (men) were obese (2/165, 1.21%), and only 4 were underweight (3 females and 1 male).

Principal component analysis of the baseline data showed that normal weight (NW) moderately correlated with the 18–30 and 31–40 age groups (r = 0.689, r = 0.760, *p* > 0.05), DW > 8 h (r = 0.740, *p* > 0.05), and physical work (r = 0.749, *p* > 0.05). Both age groups were substantially correlated with DW > 8 h and physical work (r = 0.996–0.999, *p* < 0.05). On the other hand, OW was strongly associated with the >50 age group and moderately correlates with the 41–50 age group (r = 0.762, *p* > 0.05) and DW = 1–4 h and 4–8 h (r = 0.778, 0.662, *p* > 0.05). Both DW periods highly correlated with the 41–50 (r =0.999–0.986, *p* < 0.05) and >50 (r = 0.943–0.873, *p* > 0.05) age groups. A remarkable correlation existed between OW and DC > 3500 (r = 0.999, *p* < 0.05) and a moderate one between OW and DC = 1000–1500, DC = 3001–3500, and intermittent fasting (r = 0.770–0.667, *p* > 0.05). Contrarily, NW correlated well with DC = 2501–3000 (r = 0.816, *p* > 0.05) and was moderately associated with DC = 2501–3000 and DC = 2001–2500 (r = 0.762, r = 0.703, *p* > 0.05) and with all daily meal frequencies (r = 0.697–0.623, *p* > 0.05).

Normal weight was substantially associated with females (r = 0.987, *p* < 0.05), while OW displayed a strong correlation with DP <50 g (r = 0.999, *p* < 0.05). NW had a good correlation with high school education (r = 0.863, *p* > 0.05), while OW correlated with males (r = 0.837, *p* > 0.05). NW moderately correlated with DP = 151–200 g, DP = 201–250 g, and master’s/doc (r = 0.789–0.709, *p* > 0.05), while OW correlated with DP = 101–150 g, DP > 250 g, and university (r = 0.728–0.633, *p* > 0.05).

### 3.4. Nutritional Supplements and Training Data

Of 165 gym goers, 38 (23.03%) were CSUs, 37 (22.42%) were LcSUs, 42 (25.45%) were PSUs, and 48 (29.09%) were PCLcSUs (Table 4).

The gym goers’ period of regular practices varied between <1 month (47/165) and >1 year (89/165), while the gym goers’ weekly training frequencies were <3 (57/165) times and ≥5 times (38/165) (*p* < 0.05); the training time ranged from <1 h (36/165) and >2 h (3/165) (*p* < 0.05) (Table 3).

Three types of exercises were available, cardio (15/165), force (89/165), and cardio + force (61/165), and the main reasons for training were muscle mass growth (MMG, 82/165), muscle mass tonus (55/165), weight loss (19/165), and competition (9/165) (*p* < 0.05) (Table 4).

Numerous NSU participants had been practicing recreationally at the gym for over one year: 18/38 CSUs, 18/27 LcSUs, 22/42 PCLcSUs, and 31/48 PSUs. A significant percentage of them started going to the gym very recently, for less than 1 month: 41.57% of PCLcSUs, 31.58% of CSUs, and 29.73% of LcSUs (Table 3).

Most PSUs went to the gym 3–4 times/week (26/42, 61.9%), followed by 18/38 CSUs (47.37%). The other 41.67% of PCLcSUs and 40.54% of LcSUs had less than 3 training sessions/week.

The gym training session duration was 1–2 h for most participants (70.27—83.33%) vs. others: <1 h (20.83–29.73%, *p* < 0.05) and >2 h (0–6.25%, *p* < 0.05).

Creatine and L-carnitine were consumed in similar percentages for force training (65.79 and 62.16%, respectively, *p* > 0.05), while PCLcS and PS were used in cardio + force and force training in the same measure (42.86–47.92%, *p* > 0.05).

Muscle mass growth was the principal training scope for all NSUs (40.54–57.89%), followed by muscular tonus (27.08–40.54%) (*p* > 0.05) (Table 3). For weight loss, LcS and PS were used in equal measure (almost 16%), while PCLcS was also most consumed for competition (10.42%).

The principal component analysis supported our results (Figure 1).

Figure 1A shows that PSUs substantially correlated with gym periods of 7–12 months and >1 year (r = 0.999, r = 0.952, *p* < 0.05), and gym frequency of 3–4 times/week (r = 0.923, *p* > 0.05). PCLcSUs considerably correlated with ≥5 times/week (r = 0.968, *p* < 0.05) and showed a good correlation with a gym period <1 month and frequency <3 times/week (r = 0.840, r = 0.838, *p* > 0.05). LcSUs and CSUs were moderately correlated with a gym period of 1–6 months (r = 0.577, *p* > 0.05).

Figure 1B highlights the substantial correlation between PCLcSUs and competition as a training scope (r = 0.968, *p* < 0.05) and cardio exercises and TS-MM tonus (r = 0.999, *p* < 0.05). Cardio exercises and TS-MM tonus strongly correlated with LcSUs (r = 0.870, *p* > 0.05), while cardio + force correlated with TS competition (r = 0.834, *p* > 0.05). CSUs moderately correlated with force exercises, PCLcSUs with cardio + force exercises and TS-MMG, while PSUs and cardio exercises correlated with TS-weight loss and cardio + force exercises with TS-MMG (r = 0.604–0.788, *p* > 0.05).

### 3.5. Specific Aspects of Protein and Amino Acid Supplement Consumption

#### 3.5.1. One-Only Supplement Consumers

Table 4 registers our findings exclusively correlated with NS consumption (period, time, frequency, dose, main reason for consumption, and side effects) for one-only NS users (117 gym goers).

Over 50% of CSUs consumed it always (22/38), while other NSs were consumed in MMG time (18/37 LcSUs and 29/37 PSUs) (*p* < 0.05). Most CSUs (22/37) and LcSUs (18/38) consumed those NSs for <1 year, while PCUs (19/42) reported 1–3 years. CS and PS were mainly consumed daily (26/38 vs. 23/42, Table 4), while LcS was principally consumed on training days (20/38). CSUs used 1–5 g (23/38) and 6–10 g (15/38) (*p* < 0.05). Most LcSUs (17/38) used 2 g LcS, while 14/38 used 1 g; only 6/38 used 6 g/daily. Mainly, PS doses were 40 g (19/42), 20 g (18/42), and 60 g (5/42). LcS was used exclusively for fat burning (37/37), PS was used for MMG (29/42) and weight loss (12/42), while CS was used for MMG (18/38) and physical effort capacity (18/38) (*p* < 0.05).

Most NS users (77/117) declared no side effects (Table 5). However, 40/117 participants declared adverse effects: 8/42 PSUs declared liver damage, 6/42 revealed muscle cramps, and 1/42 had kidney damage. In LcUs, 9/38 mentioned nausea and 5/38 had stomach cramps. CSUs claimed liver damage (6/37) and weight gain (3/37) (*p* < 0.05) (Table 5).

The principal component analysis supported the data from Table 4, which evidences the correlations between variable parameters (Figure 2).

Figure 2A shows a significant correlation between CSUs and physical effort capacity (r = 0.999, *p* < 0.05), PSUs and physical effort recovery and weight loss (r = 0.999, *p* < 0.05), and LcSUs and fat-burning and consumption in MD time (r = 0.999, *p* < 0.05).

Figure 2B shows that diarrhea, nausea, and stomach cramps were substantially associated with LcS at 1–2 g/dose (r = 0.999, *p* < 0.05). PS at 20, 40, and 60 g/dose significantly correlated with muscle cramps and liver damage (r = 0.999, *p* < 0.05), while CS at 1–5 g strongly correlated with weight gain (r = 0.999, *p* < 0.05). CS also correlated with 6–10 g/dose and kidney damage (r = 0.918–0.988, *p* > 0.05).

The chi-square independence test assessed the significance of each variable parameter’s influence on the risk of potential side effects; all *p*-values are reported in Table 6.

Therefore, NS type and NS dose were the primary influencing factors of the potential risk of harmful effects (*p* < 0.0001), followed by body weight status (*p* = 0.001).

#### 3.5.2. Combination (PCLcS) Consumers

All 48 participants consumed all 3 NSs daily to obtain maximal benefits (Table 7).

They mainly consumed PS always (28/48), LcS in MD time (36/48), and CS in MMG (36/48). They used all NSs for almost <1 year (PS, 22/48; LcS, 18/48, and CS, 27/48, *p* < 0.05). CS and PS were used daily (35/48 and 25/48, *p* < 0.05), while LcS was only used on training days (47/48) in almost the same doses as the one-only NSUs (recorded in Table 4). PS and CS were primarily used for MMG (31/48 and 34/48) and LcS for fat burning (45/48). All NSs were also used for physical effort recovery (3/48 used PS and LcS, while 6/48 used CS), 14/48 was used PS for weight loss, and 8/48 used CS for physical effort capacity (*p* < 0.05, Table 7).

All PCLcSUs reported side effects of each component: muscle cramps and liver damage were evidenced by 15/48 after PS consumption, while 10/48 claimed kidney damage and weight gain caused by CS use (*p* < 0.05). LcS administration’s side effects were diarrhea, nausea, stomach cramps, and vomiting, which were experienced by 23/48 gym goers (*p* < 0.05, Table 7).

### 3.6. Protein and Amino Acid Supplement Consumption—Significant Correlations with Baseline Data

#### 3.6.1. NS Consumption and Socio-Demographic Data

Protein and amino acid supplementation were influenced by socio-demographic factors (sex, age group, educational level, monthly income, daily working time, and work type; Figure 3A–C). Females preferred PS and CS to PCLcS and LcS (38.2% and 34.5% vs. 14.5% and 12.7%, *p* < 0.05, Figure 3A). The most-used NSs by males were PCLcS (36.4%) and LcS (27.3%). Their preference for PS and CS was significantly diminished (19.1% and 17.3%, *p* < 0.05, Figure 3A). Recreational gym goers aged 18–30 mostly preferred PCLcS (39.1%). The use of this combination significantly decreased with age progression (31–40 and 41–50: 20.5% and 17.1%, respectively, *p* < 0.05); the NS combination was not used by recreational gym goers aged 51–60 (Figure 3A). CSU’s percentage of use increased proportionally with age: 18–30 vs. 31–40 vs. 41–50 = 19.5 vs. 25.6 vs. 31.4 (*p* < 0.05); however, the 50–60 age group did not use CS (Figure 3A). Protein consumption was similar across all age groups (25–25.7%), while L-carnitine use varied significantly, ranging from 19.5% in the 18–29 age group to 75% in the 50–60 age group of participants (*p* < 0.05, Figure 3A).

The NS preferences of high school gym goers and those with a monthly income of 2000–4000 RON were not significantly different (*p* > 0.05, Figure 3B). The PCLcS, CS, and LcS consumption significantly differed between graduate and postgraduate participants: 33.3%, 25.7%, and 16.2% vs. 20%, 13.3%, and 40%, respectively, *p* < 0.05, Figure 3B). The preference for PCLcS slightly increased with monthly income, from 2000–3000 RON to over 6000 RON, while LcS use remained similar (*p* > 0.05, Figure 3B).

Creatine (*p* < 0.05) and protein consumption (*p* > 0.05) decreased with work type, from office to physical work. Protein consumption increased with DW hours, while creatine consumption diminished with them (*p* < 0.05, Figure 3C).

The Pearson’s correlations supported our findings. The correlation coefficient (r) showed that CSU significantly correlated with office work, while LcSU showed a substantial association with master’s/doctorate (r = 0.992–0.997, *p* < 0.05). PCLcSU highly correlated with males and university educational level, DW = 4–8 h and >8 h (r = 0.967–0.999, *p* < 0.05). Moreover, PS consumption was strongly associated with office work (r = 0.992, *p* < 0.05), while PCLcS was related to physical work (r = 0.988, *p* < 0.05). CS consumption substantially correlated with the monthly income (MI) of 4000–6000 RON. PS and PCLcS were significantly associated with MI >6000 RON (r = 0.970–0.996, *p* < 0.05), while PCLcS also strongly correlated with MI <2000 RON (r =0.974, *p* < 0.05).

#### 3.6.2. NS Consumption, Daily Meal Frequency, and Unhealthy Habits

PCLcS was consumed by gym goers with 2 meals and those with 3 meals and 2 snacks, higher than another NS (*p* < 0.05), while PS and CS were the leading choices for those with IF and 3 meals (*p* > 0.05, Figure 4A). No significant differences were recorded in LcS preferences (*p* > 0.05, Figure 4A).

No significant differences were found between non-smoker gym goers and NS consumption (22.4–26.7%, *p* > 0.05). PCLcS was mainly preferred by smokers (34.7%), while all others were less preferred (*p* < 0.05, Figure 4B). The same preferences were shown by gym goers who consumed alcohol; PCLcS was the first choice (*p* > 0.05, Figure 4B). Non-alcohol consumers most frequently chose protein and creatine (Figure 4B).

#### 3.6.3. NS Consumption, Body Weight Status, and Daily Diet Properties

Obese gym goers consumed only L-carnitine (100%), while underweight ones chose creatine (75%) and L-carnitine (25%) (*p* < 0.05, Figure 5A). Overweight and normal-weight participants mainly preferred PCLcS and protein (Figure 5A). Both supplements were primarily chosen for participants with balanced and low-carb diets. Those with a hyperprotein diet preferred L-carnitine, while vegetarians used creatine and protein supplements (Figure 5A).

The principal component analysis showed that LcS consumption substantially correlated with obese gym goers (r = 0.999, *p* < 0.05). By contrast, CS administration strongly correlated with UW (r = 0.943, *p* > 0.05), PS with NW (r = 0.889, *p* > 0.05), and PCLcS with OW (r = 0.898, *p* > 0.05).

PCLcS and PS consumption showed a considerable association with the low-carb daily diet (DD), LcS and PCS highly correlated with a balanced DD, and PCLcS use was also strongly associated with a vegetarian diet (r = 0.960–0.986, *p* < 0.05).

Participants with >3500 calories and >250 g proteins used PCLcS exclusively, while those with <50 g proteins completed their diet with protein supplementation (Figure 5B,C). These supplements were the leading choice for gym goers with 3001–3500 calories and 1000–1500 calories (PCLcS, *p* < 0.05) and 50–100 g protein (PS, *p* < 0.05).

## 4. Discussion

The consumption of nutritional supplements among exercising individuals has increased unnecessarily. Adequate research is essential to clarify the various facts regarding the necessity, efficacy, and appropriate use of dietary supplements. In a survey of 1120 gym goers in Brazil, 36.8% reported regularly taking supplements such as protein and creatine to build muscle and strength. Products high in protein and amino acids were consumed nearly every day by almost 60% of the participants, followed by isotonic beverages and carbohydrates, with percentages of 32% and 23%, respectively. Supplements high in protein were taken by those under 30 years old, mostly men [123]. This observation also fits our findings. The present study enrolled 165 participants (men to women ratio = 2:1) who were recreational gym practitioners. Their daily diet included an NS (protein, creatine, L-carnitine) or a combination of these as a routine part of their lifestyle. The cohort was analyzed according to the type of NS intake. The PSU group accounted for almost 23%, while the CSU and LcSU groups comprised 22% and 25%, respectively, and the PCLcSU group was 29%. The socio-demographic data for the entire cohort revealed that most participants were male (66%), compared to approximately 33% female. Over 52% were aged 18–30; thus, our findings confirm the young gym goer profile [124]. The outcomes for professional activity demonstrated that the cohort was balanced among individuals with office work, physical activity, and mixed activity that combined office work and physical effort. According to their BMI, nearly 50% were of normal weight, 46% were overweight, 1% were obese, and 2% were underweight. These low percentages for extreme BMI values suggest that our gym goers prioritized their health and physical appearance. Our study revealed significant associations between occupational activity, age, gender, and body weight categories. Obese participants had office work, suggesting that sedentary lifestyles in office-based occupations significantly contribute to excessive weight gain. The findings align with the existing literature indicating that reduced physical activity in workplace settings is a significant factor contributing to the rising prevalence of obesity [125].

In the present study, dietary patterns showed substantial correlations with body weight and gender (*p* < 0.05). Almost 65% of gym goers had a balanced diet, while 19% opted for a hyperprotein diet (*p* < 0.05). Vegetarian and low-carb diets were rare (9% and 6%, *p* < 0.05). Sedentary participants were predominantly oriented toward a vegetarian diet (14.5%), while obese individuals preferred a balanced diet. Participants with a vegetarian diet, with high-calorie (DC >3500) and low protein (DP <50 g), were highly associated with overweight status (*p* < 0.05). Many gym goers used protein supplements to reach this daily amount. However, 17% of participants were unable to provide their daily protein intake (14%) and daily calorie intake (3%) due to a lack of nutritional knowledge, as previously reported [126]. Moreover, balanced and vegetarian diets were the most affordable, with 50% and 62.5% of participants, respectively, having a monthly income of 2000–4000 RON.

Our findings reported that a balanced diet consisted of 1501–2500 calories per day (68%) and 50–150 g of protein (54%). In this context, almost 59% of recreational gym goers were PCLcSUs and PSUs, even if the extra supplementation was unnecessary; our findings fit the literature data [77]. The hyperprotein diet involved a daily protein intake of 151–250 g (87.5%) and a daily calorie consumption of 2501–3000 kcal (90.6%), while the most used nutrient supplement was L-carnitine (37.5%, *p* < 0.05). Office workers preferred a low-carb diet (60%); they consumed 1000–2000 calories/daily (80%) and 151–250 g of protein (75%). A vegetarian diet also involved 1000–2000 calories per day (93%) but less proteins (50–150 g/day, 62.5%). LcS and PS highly correlated with a balanced diet, while PCLcS use was strongly associated with a vegetarian diet (r = 0.960–0.986, *p* < 0.05). Hyperprotein and low-carb diets required higher incomes: 4001–6000 RON (43.75%) and over 6000 RON (40%), respectively. Moreover, bad habits (alcohol consumption and smoking) increased with monthly expenses (monthly income >4000 RON: 55.54% and 67.35%, respectively), while most consumers (>50%) were young (18–30 years old) and preferred the triple combination (PCLcS). Associating protein, creatine, and L-carnitine (29%) and consuming them on the same day has synergistic benefits in muscle growth, strength, and recovery [44,127,128].

The claims of increased muscle mass, higher fat loss, improved performance, and quick recovery lead to the consumption of protein and amino acid supplements. The present study’s results showed that the single choice of obese gym-goers was L-carnitine, known as a fat-burner [129], while underweight individuals chose creatine and L-carnitine (*p* < 0.05). Creatine induces weight gain [130], while L-carnitine may improve energy and physical activity, which could indirectly support weight gain if combined with a proper diet and strength training [131].

Correlating the training period (in months), frequency (times per week), duration (in hours), and NS consumption, the outcomes highlighted the complex interplay between occupational activity, dietary habits, supplementation, training patterns, and body weight, emphasizing the need for targeted interventions to promote healthier behavioral choices. The study revealed strong associations between NS use and training duration: the PCLcS combination was nearly perfectly associated with training sessions exceeding 2 h and a frequency of more than 5 times per week (*p* < 0.05), while PS substantially correlated with a gym period of more than 1 year (*p* < 0.05). These findings confirm previous data regarding their benefits in RT [132,133].

Our gym goers similarly used CS and LcS to ensure performance in force training (65.79% vs. 62.16%). Only CS use benefits are confirmed by literature data [23,134,135,136]. Moreover, they did not optimally valorize the strong and verified effects of PS or the synergistic combination of PCLcS (42.86% and 47.92%, respectively). These findings suggest that the participants selected NSs themselves and consumed protein supplements without the recommendation of a qualified practitioner, as previous studies claimed [126,137]. Only PCLcS was correctly associated with competitive scope (*p* < 0.05), demonstrating the correct supplementation that is professionally supervised for potential elite athletes.

Moreover, in one-only supplement consumption, there were significant correlations between CSUs and physical effort capacity, PSUs and physical effort recovery and weight loss, and LcSUs and fat-burning and consumption in MD time (*p* < 0.05).

In addition, the high incidence of harmful effects claimed by NS users (30% in one-only NS use and 100% in triple combination) supports the previous warnings regarding progressively increased and unnecessary sports supplement consumption [77,138,139]. Diarrhea, nausea, and stomach cramps were substantially associated with LcS at 1–2 g/dose (*p* < 0.05). Whey protein at 40 and 60 g/doses significantly correlated with muscle cramps and liver damage (*p* < 0.05), while CS at 1–5 g strongly correlated with weight gain (*p* < 0.05). NS type and daily dose were the primary factors associated with the risk of potential side effects reported by participants (*p* < 0.0001).

### Strengths and Limitations of the Present Study

The present study offers complex data regarding protein and amino acid supplements consumed by recreational gym goers, considering various influential factors. Two qualified coaches selected the participants based on direct discussions with regular gym members they had met during their training sessions. Moreover, data collected through face-to-face interviews were more accurate than those obtained through an online questionnaire.

The results of the present study confirmed all hypotheses. Most recreational gym goers were young (18–30 years old) and had completed academic studies. The men’s percentage was twice that of women’s; of the total group, 25% consumed all NSs in combination. Our findings indicate that supplement consumption is influenced by sex, age, body weight, diet, gym duration, and the type and scope of training. The present study also revealed that all components act synergistically in the triple combination, stimulating muscle growth and strength. Furthermore, the risk for potential side effects is significantly influenced by the NS type and dose.

Even if our study contains valuable insights, several limitations should be acknowledged, as follows:The cohort selection process, which included only the regular members of 2 gym centers from the same Romanian city, did not ensure an optimal representation of all age groups;All data were obtained from the participants, guaranteed by self-responsibility;The cross-sectional design limits the possibility of establishing relationships between cause and effect;The absence of measurements of biological parameters restricts the accurate evaluation of the risk for potential harmful effects of protein and amino acid supplements on health.

## 5. Conclusions

The progressive increase in the self-administration of nutritional supplements poses a challenge to control and diminish, as recreational gym goers are also adopting them.

Aiming to fill the gap between consumers’ beliefs, expectations, and the potentially harmful effects of these supplements, the present study thoroughly investigated the consumption of commonly used protein and amino acids by recreational gym goers in Northwestern Romania. It provides essential data about this phenomenon, which is associated with numerous factors, revealing that the type of NS and daily dose substantially influences the potential risk for harmful effects.

Our findings underscore the need for educational programs that focus on healthy nutrition, ensuring that the necessary nutrients are primarily obtained through an optimal diet. Therefore, the excessive consumption of high doses of protein and amino acid supplements without guidance from a healthcare professional may be reduced.

## Figures and Tables

**Figure 1 jfmk-10-00248-f001:**
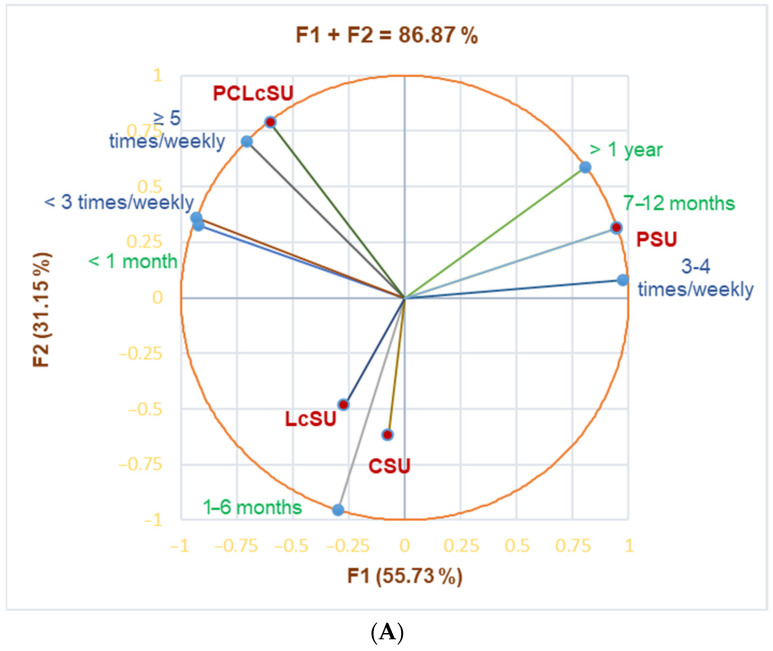
Correlations between NS consumption and (**A**) training period (months) and frequency (times/weekly); (**B**) training type and scope. LcSU—L-carnitine supplement user, CSU—creatine supplement user, PSU—protein supplement user, PCLcSU—protein + creatine + L-carnitine supplement user. MM—muscle mass, TS—training scope.

**Figure 2 jfmk-10-00248-f002:**
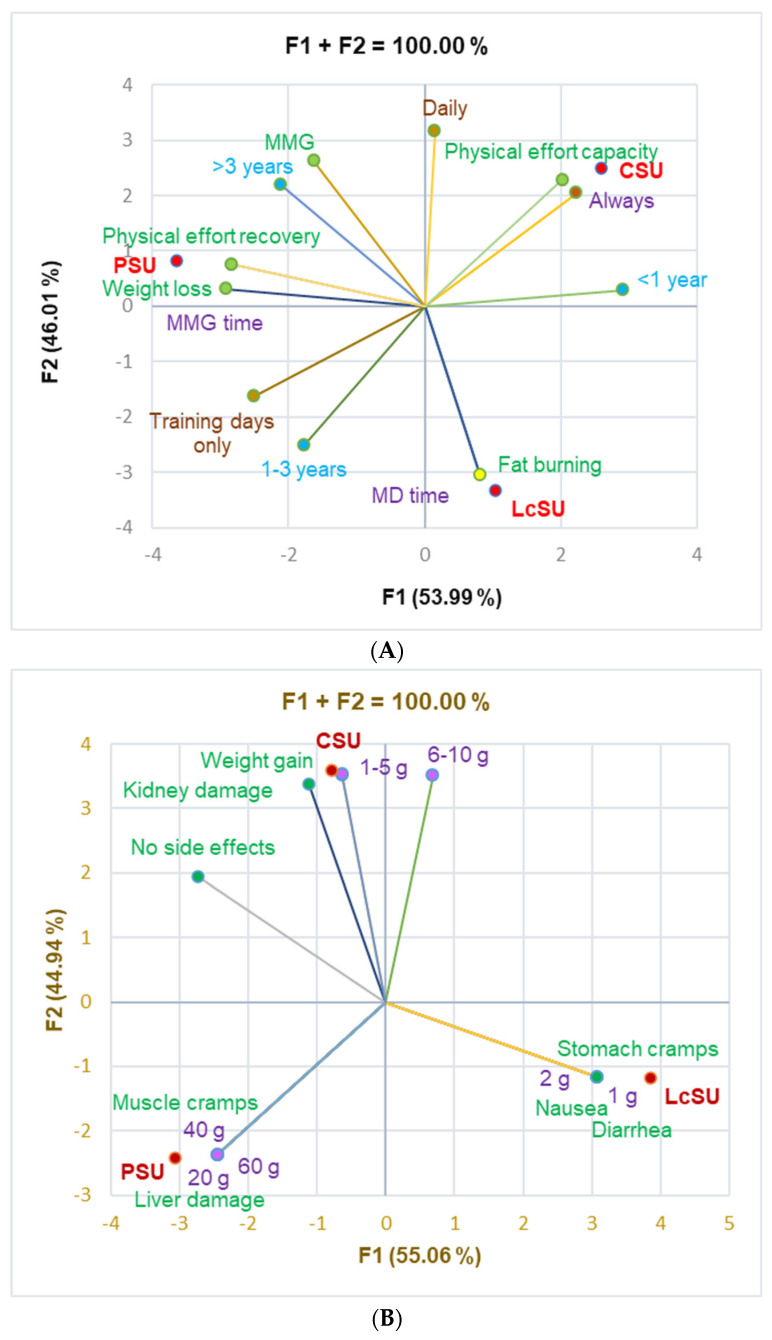
Correlations between specific aspects of one-only supplement users: (**A**) period, time, and the main reason for consumption; (**B**) dose and side effects.

**Figure 3 jfmk-10-00248-f003:**
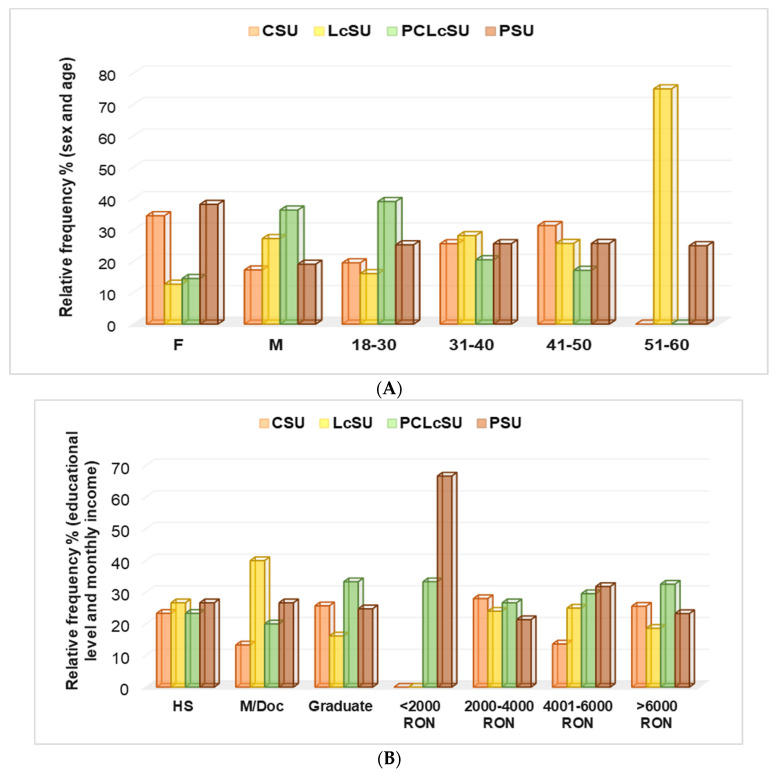
NS consumption and socio-demographic patterns: (**A**) sex and age; (**B**) educational level and monthly income; (**C**) work type and daily working time. F—females, M—males, HS—high school, Graduate—university, M/Doc—master’s/doctorate, PW—physical work; DW—daily working hours.

**Figure 4 jfmk-10-00248-f004:**
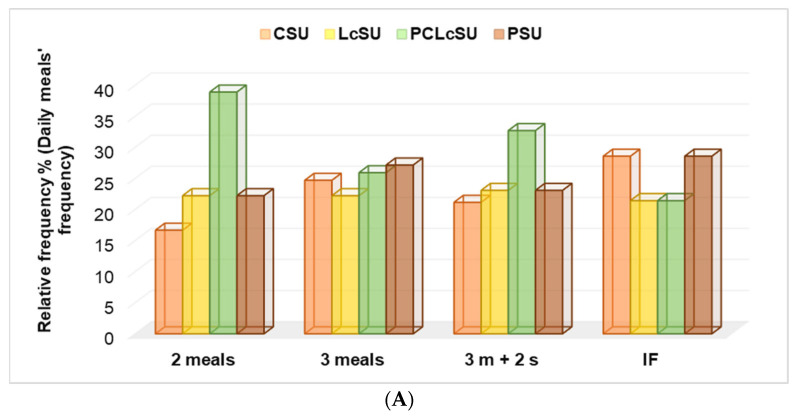
NS consumption and (**A**) daily meals, (**B**) unhealthy habits. LcS—L-carnitine supplement, CS—creatine supplement, PS—protein supplement, PCLcS—protein + creatine + L-carnitine supplement, U—user, IF—intermittent fasting, 3 m + 2 s—3 meals + 2 snacks, SS—smoking status, AC—alcohol consumption.

**Figure 5 jfmk-10-00248-f005:**
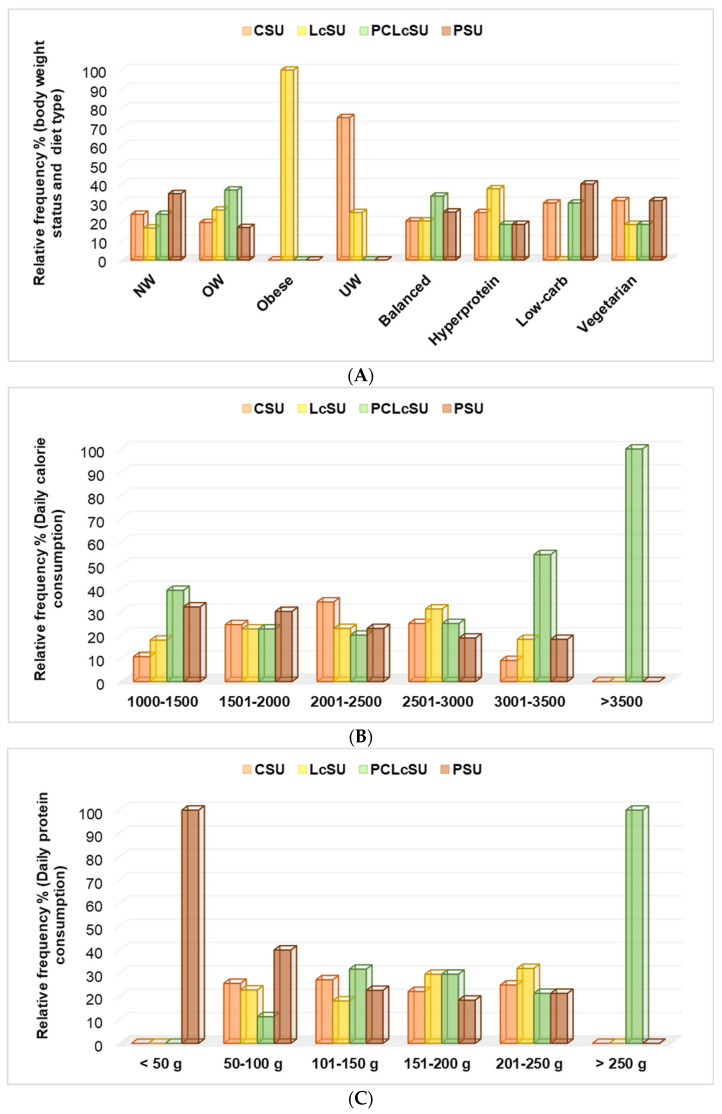
NS consumption and (**A**) body weight and diet type; (**B**) daily calorie consumption; (**C**) daily protein consumption. LcS—L-carnitine supplement, CS—creatine supplement, PS—protein supplement, PCLcS—protein + creatine + L-carnitine supplement, U—user, NW—normal weight, OW—overweight; UW—underweight.

**Table 1 jfmk-10-00248-t001:** Inclusion and exclusion criteria in the study cohort.

Inclusion Criteria	Exclusion Criteria
Age	
- 18–60 years old	- <18 years and >60 years
Health Status and Particular Conditions	
- Healthy individuals (males and females).	- Individuals with severe illnesses (musculoskeletal disorders, cancer, liver disease, heart failure, or kidney failure).
Gym Goers Status	
- Recreational gym goers, members of the above-mentioned gym centers, with regular training sessions at least twice a week.	- Members of the gym centers mentioned above who had fewer than 2 training sessions in a week; - Individuals who commonly perform their gym training at other gym centers.
NS Consumption	
- Protein and amino acid consumers in various formulations for at least 6 months.	- Individuals who use androgenic steroids, diuretics, epinephrine, and other prohibited substances in the gym that are interdicted by the World Anti-Doping Agency (WADA) [115]; - Other NS consumers; - Non-NS users.

NS—nutritional supplement. Data regarding nutritional supplement consumption was provided by each gym goer on their own responsibility.

**Table 2 jfmk-10-00248-t002:** Demographic and socio-economic data of the study participants.

Parameter	Total Participants		F		M		*p*-Value
	*n*	%	*n*	%	*n*	%	
Total	165	100	55	33.33	110	66.67	<0.05
Age (years)							
18–30	87	52.73	24	43.64	63	57.27	<0.05
31–40	39	23.64	18	32.73	21	19.09	<0.05
41–50	35	21.21	10	18.18	25	22.73	<0.05
51–60	4	2.42	3	5.45	1	0.91	<0.05
Education							
High School	30	18.18	11	20.00	19	17.27	<0.05
Master’s/Doc	30	18.18	9	16.36	21	19.09	<0.05
Graduate	105	63.64	35	63.64	70	63.64	<0.05
Monthly Income (RON)							
<2000	3	1.82	2	3.64	1	0.91	<0.05
2000—4000	75	45.45	28	50.91	47	42.73	<0.05
4001—6000	44	26.67	12	21.82	32	29.09	<0.05
>6000	43	26.06	13	23.64	30	27.27	<0.05
Daily Working Period (hours)							
1–4 h	5	3.03	1	1.82	4	3.64	<0.05
4–8 h	69	41.82	25	45.45	44	40.00	<0.05
>8 h	91	55.15	29	52.73	62	56.36	<0.05
Work Type							
Office	54	32.73	19	34.55	35	31.82	<0.05
Physical work	44	26.67	15	27.27	29	26.36	<0.05
Physical work + office	67	40.61	21	38.18	46	41.82	<0.05

*n*—number of participants (frequency), %—percentage (relative frequency), F—female, M—male. Statistical significance: *p* < 0.05.

**Table 3 jfmk-10-00248-t003:** Lifestyle-related data of participants.

Parameter	Total Participants		F		M		*p*-Value
	*n*	%	*n*	%	*n*	%	
Smoking Status							
SS No	116	70.3	40	72.73	76	69.09	<0.05
SS Yes	49	29.7	15	27.27	34	30.91	<0.05
Alcohol Consumption							
AC No	32	19.39	16	29.09	16	14.55	>0.05
AC Yes	133	80.61	39	70.91	94	85.45	<0.05
Daily Diet Type							
Balanced	107	64.85	30	54.55	77	70	<0.05
Hyperprotein	32	19.39	13	23.64	19	17.27	<0.05
Low-carb	10	6.06	3	5.45	7	6.36	<0.05
Vegetarian	16	9.7	9	16.36	7	6.36	<0.05
Daily Meals							
2 meals	18	10.91	8	14.55	10	9.09	<0.05
3 meals	81	49.09	23	41.82	58	52.73	<0.05
3 meals + 2 snacks	52	31.52	19	34.55	33	30	<0.05
Intermittent fasting	14	8.48	5	9.09	9	8.18	<0.05
Daily Calorie Consumption							
NA	5	3.03	1	1.82	4	3.64	<0.05
>3500	1	0.61	0	0.00	1	0.91	<0.05
1000—1500	28	16.97	13	23.64	15	13.64	<0.05
1501—2000	53	32.12	15	27.27	38	34.55	<0.05
2001—2500	35	21.21	12	21.82	23	20.91	<0.05
2501—3000	32	19.39	11	20.00	21	19.09	<0.05
3001—3500	11	6.67	3	5.45	8	7.27	<0.05
Daily Protein Consumption (g)							
NA	23	13.94	5	9.09	18	16.36	<0.05
<50 g	1	0.61	1	1.82	0	0	<0.05
101—150 g	44	26.67	13	23.64	31	28.18	<0.05
151—200 g	27	16.36	11	20.00	16	14.55	<0.05
201—250 g	28	16.97	8	14.55	20	18.18	<0.05
50—100 g	35	21.21	15	27.27	20	18.18	<0.05
>250 g	7	4.24	2	3.64	5	4.55	<0.05
Body Weight Status							
NW	83	50.3	44	80.00	39	35.45	<0.05
Obese	2	1.21	0	0.00	2	1.82	<0.05
OW	76	46.06	8	14.55	68	61.82	<0.05
UW	4	2.42	3	5.45	1	0.91	<0.05

*n*—number of participants (frequency), %—percentage (relative frequency), F—female, M—male, NA—no available data (the participants responded “I don’t know” to the corresponding question); AC—alcohol consumption; SS—smoking status; NW—normal weight, OW—overweight; UW—underweight. Statistical significance: *p* < 0.05.

**Table 4 jfmk-10-00248-t004:** Gym training characteristics associated with the consumption of nutritional supplements.

Parameter	CSU		LcSU		PCLcSU		PSU	
	*n*	%	*n*	%	*n*	%	*n*	%
Total	38	23.03	37	22.42	48	29.09	42	25.45
Gym Period								
1–6 months	8	21.05	8	21.62	6	12.5	6	14.29
7–12 months	0	0	0	0.00	0	0	1	2.38
<1 month	12	31.58	11	29.73	20	41.67	4	9.52
>1 year	18	47.37	18	48.65	22	45.83	31	73.81
Gym Training Weekly Frequency								
3–4 times/week	18	47.37	13	35.14	13	27.08	26	61.9
<3 times/week	13	34.21	15	40.54	20	41.67	9	21.43
≥5 times/week	7	18.42	9	24.32	15	31.25	7	16.67
Gym Training Session								
1–2 h	30	78.95	26	70.27	35	72.92	35	83.33
<1 h	8	21.05	11	29.73	10	20.83	7	16.67
>2 h	0	0	0	0	3	6.25	0	0
Training Type								
Cardio	3	7.89	5	13.51	3	6.25	4	9.52
Cardio + Force	10	26.32	9	24.32	22	45.83	20	47.62
Force	25	65.79	23	62.16	23	47.92	18	42.86
Training Scope (TS)								
Competition	1	2.63	1	2.7	5	10.42	2	4.76
MM Tonus	13	34.21	15	40.54	13	27.08	14	33.33
MMG	22	57.89	15	40.54	26	54.17	19	45.24
Weight loss	2	5.26	6	16.22	4	8.33	7	16.67

*n*—number of participants (frequency), %—percentage (relative frequency), LcSU—L-carnitine supplement user, CSU—creatine supplement user, PSU—protein supplement user, PCLcSU—protein + creatine + L-carnitine supplement user, MM—muscle mass, MMG—muscle mass growth.

**Table 5 jfmk-10-00248-t005:** Specific aspects of NS consumption in one-only nutritional supplement users (CSUs, LcSUs, PSUs).

Parameter	CS		LcS		PS		*p*-Value
	*n*	%	*n*	%	*n*	%	
NS Use Time							
Always	22	57.89	14	37.84	13	30.95	<0.05
MD time	0	0	5	13.51	0	0	<0.05
MMG time	16	42.11	18	48.65	29	69.05	<0.05
NS Use Duration (years)							
1–3 years	8	21.05	16	43.24	19	45.24	<0.05
<1 year	22	57.89	18	48.65	10	23.81	<0.05
>3 years	8	21.05	3	8.11	13	30.95	<0.05
NS Consumption Frequency							
Daily	26	68.42	17	45.95	23	54.76	<0.05
Training days only	12	31.58	20	54.05	19	45.24	<0.05
NS Daily Dose							
1 g	0	0	14	37.84	0	0	<0.05
1–5 g	23	60.53	0	0	0	0	<0.05
2 g	0	0	17	45.95	0	0	<0.05
6–10 g	15	39.47	6	16.22	0	0	<0.05
20 g	0	0	0	0	18	42.86	<0.05
40 g	0	0	0	0	19	45.24	<0.05
60 g	0	0	0	0	5	11.9	<0.05
The Main Reason for NS Consumption							
Fat burning	0	0	37	100	0	0	<0.05
MMG	20	52.63	0	0	29	69.05	<0.05
Physical effort capacity	18	47.37	0	0	0	0	<0.05
Physical effort recovery	0	0	0	0	1	2.38	<0.05
Weight loss	0	0	0	0	12	28.57	<0.05
Side Effects							
Diarrhea	0	0	2	5.41	0	0	<0.05
Kidney damage	6	15.79	0	0	1	2.38	<0.05
Liver damage	0	0	0	0	8	19.05	<0.05
Muscle cramps	0	0	0	0	6	14.29	<0.05
Nausea	0	0	9	24.32	0	0	<0.05
No side effects	29	76.32	21	56.76	27	64.29	<0.05
Stomach cramps	0	0	5	13.51	0	0	<0.05
Weight gain	3	7.89	0	0	0	0	<0.05

*n*—number of participants (frequency), %—percentage (relative frequency), LcS—L-carnitine supplement, CS—Creatine supplement, PS—protein supplement, LcSUs—L-carnitine supplement users, CSUs—creatine supplement users, PSUs—protein supplement users, MMG—muscle mass growth, MD—muscle defining.

**Table 6 jfmk-10-00248-t006:** Chi-square independence test results.

Variable Parameters	*p*-Value
NS-type	<0.0001
NS dose	<0.0001
Body weight status	0.001
NS use time	0.061
NS duration	0.209
Sex	0.233
Smoking status	0.287
AC frequency	0.291
Daily protein consumption	0.358
Daily meal frequency	0.823
Daily calorie consumption	0.825
Age	0.864
NS consumption frequency	0.869
Activity type	0.878
Daily Diet Type	0.979

NS—nutritional supplement, AC—alcohol consumption.

**Table 7 jfmk-10-00248-t007:** Specific aspects of NS consumption in PCLcSUs.

Parameter	PS		LcS		CS		*p*-Value
	*n*	%	*n*	%	*n*	%	
Use Time							
Always	28	58.33	6	12.5	12	25	<0.05
MMG time	20	41.67	6	12.5	36	75	<0.05
MD time	0	0	36	75	0	0	<0.05
Use Duration (years)							
1–3 years	12	25.00	15	31.25	11	22.92	<0.05
<1 year	22	45.83	18	37.5	27	56.25	<0.05
>3 years	14	29.17	15	31.25	10	20.83	<0.05
NS Consumption Frequency							
Daily	25	52.08	1	2.08	35	72.92	<0.05
Training days only	23	47.92	47	97.92	13	27.08	<0.05
Dose							
20 g	25	52.08	0	0	0	0	<0.05
40 g	20	41.67	0	0	0	0	<0.05
60 g	3	6.25	0	0	0	0	<0.05
1 g	0	0	6	12.5	0	0	<0.05
2 g	0	0	8	16.67	0	0	<0.05
3 g	0	0	2	4.17	0	0	<0.05
1–5 g	0	0	0	0	38	79.17	<0.05
6–10 g	0	0	32	66.67	10	20.83	<0.05
The Main Reason for Consumption							
MMG	31	64.58	0	0	34	70.83	<0.05
Physical effort recovery	3	6.25	3	6.25	6	12.5	<0.05
Weight loss	14	29.17	0	0	0	0	<0.05
Fat burning	0	0	45	93.75	0	0	<0.05
Physical effort capacity	0	0	0	0	8	16.67	<0.05
Side Effects							
Liver damage	6	12.5	0	0	0	0	<0.05
Muscle cramps	9	18.75	0	0	0	0	<0.05
Diarrhea	0	0	2	4.17	0	0	<0.05
Nausea	0	0	9	18.75	0	0	<0.05
Stomach cramps	0	0	8	16.67	0	0	<0.05
Vomiting	0	0	4	8.33	0	0	<0.05
Kidney damage	0	0	0	0	5	10.42	<0.05
Weight gain	0	0	0	0	5	10.42	<0.05

*n*—number of participants (frequency), %—percentage (relative frequency), LcS—L-carnitine supplement, CS—creatine supplement, PS—protein supplement, PCLcSUs—protein + creatine + L-carnitine supplement user, MMG—muscle mass growth, MD—muscle defining.

## Data Availability

The original contributions presented in the study are included in the article; further inquiries can be directed to the first author and corresponding authors.

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
