# Peer review of "Protein and Amino Acid Supplementation Among Recreational Gym Goers and Associated Factors—An Exploratory Study"

_jfmk, 2025, doi:10.3390/jfmk10030248_

Round 1

Reviewer 1 Report

Comments and Suggestions for Authors

The present study aimed to investigate the relationship between protein and amino acid supplementation and various associated aspects among recreational athletes.

Some aspects are unclear, the focus of the study should be explicited more precisely, as well as the population better described and the findings better explained/contextualized.

More in detail and starting from the title/abstract: are the subjects included really considerable athletes (even if recreational?)? They are engaged in any sport but normal people exercising in the gym, so far from being any kind of athlete. The etherogenicity of the population pose a clear limit to the interpretation of data, "gym-goers" may include a wide spectrum of subjects and they these subjects are engaged to their training in not only defined by the frequency but also by volume, intensity and quality of their training. Putting all together could lead to misinterpreation of the results. Already from the first sentence of the abstract the authors seem to be biased on the fact that supplement such as protein and creatine may need medical advice to be used. That is a concept not sufficiently proved by scientific rationale and this is confirmed by a lack of rationale supporting this concept/statement also in the introduction. And again, the final sentence of the abstract (conclusion) pend to this, even if the study itself does not seem to fully support this concept. The authors are invited to revise this aspect. Protein and creatine supplements are just elements normally present in the diet and their use represents just a "supplement" to normal diet with very limited harmful side effects, and not different from the effect of abusing of food sources of such elements with the diet. This is widely supported by several position stands (e.g., those of the ISSN: https://doi.org/10.1186/s12970-017-0173-z; https://doi.org/10.1186/s12970-017-0177-8.)

From the introduction of the authors is already clear that, side effects are almost always reported in relation with abuse + cohomorbities, that is not the case of the population in study since the inclusion-exclusion criteria eliminate such population at risk. Therefore, starting from the abstract to the discussion, the interpretation of their results and claims should be reconsidered in light of a healthy population at minimal risk of side effects even when relatively speaking "abusing" of these supplements.

Minor comments:

An example of the questionnaire used would benefit the revision process. I see the authors mentioned a supplementary file and I believe that could be the questionnaire but unfortunately the link does not seem to work.

The graphs in figure 3, 4, 5, could benefit of including in the Y axis explicitily the relativre frequency to which variable being linked (e.g., relative frequency (gender and age)).

Author Response

Dear Reviewer 1, 

The authors are grateful for your time and attention, as well as for your accurate and valuable comments, which aim to improve the quality of the present manuscript. The authors responded point by point to each comment, and they would be pleased to know that they succeeded. Please find their response attached. 

Reviewer 2 Report

Comments and Suggestions for Authors

 The article addresses a relevant and under-researched topic in the recreational field.
The introduction and abstract are too verbose and with too much data.

Inclusion of well-defined criteria for enrolling participants.
Statistical analysis is articulated and uses advanced tools (PCA, Cronbach's α).

Has the questionnaire already been used/validated? Is this the first time it has been used?
The statistical analysis is almost too “nice”, doubts may arise.

Improve the discussion section with more critical thinking and indicate what the limitations of the study are.

The graphs are not very immediate as they are full of data. Simplifying them might help the reading

Author Response

Dear Reviewer 2, 

The authors are grateful for your time and attention, as well as for your accurate and valuable comments, which aim to improve the quality of the present manuscript. The authors responded point by point to each comment, and they would be pleased to know that they succeeded. Please find their response attached. 
